# Efficient Architecture Search for Diverse Tasks

**Junhong Shen**[*]
Carnegie Mellon University
junhongs@andrew.cmu.edu

**Mikhail Khodak**[*]
Carnegie Mellon University
khodak@cmu.edu

**Ameet Talwalkar**
Carnegie Mellon University
talwalkar@cmu.edu

## Abstract

While neural architecture search (NAS) has enabled automated machine learning (AutoML) for well-researched areas, its application to tasks beyond computer vision is still under-explored. As less-studied domains are precisely those where we expect AutoML to have the greatest impact, in this work we study NAS for efficiently solving *diverse* problems. Seeking an approach that is fast, simple, and broadly applicable, we fix a standard convolutional network (CNN) topology and propose to search for the right kernel sizes and dilations its operations should take on. This dramatically expands the model's capacity to extract features at multiple resolutions for different types of data while only requiring search over the operation space. To overcome the efficiency challenges of naive weight-sharing in this search space, we introduce DASH, a differentiable NAS algorithm that computes the mixture-of-operations using the Fourier diagonalization of convolution, achieving both a better asymptotic complexity and an up-to-10x search time speedup in practice. We evaluate DASH on ten tasks spanning a variety of application domains such as PDE solving, protein folding, and heart disease detection. DASH outperforms state-of-the-art AutoML methods in aggregate, attaining the best-known automated performance on seven tasks. Meanwhile, on six of the ten tasks, the combined search and retraining time is less than 2x slower than simply training a CNN backbone that is far less accurate.

## 1 Introduction

The success of deep learning for computer vision and natural language processing has spurred growing interest in enabling similar breakthroughs for other domains such as biology, healthcare, and physical sciences. Consequently, there is enormous potential for neural architecture search to help automate model development in these diverse areas. However, while extensive NAS research has been devoted to improving the search speed [1, 2] and automatically attaining state-of-the-art performance on vision datasets such as CIFAR and ImageNet [3], the resulting algorithms have subpar performance beyond the tasks on which they were developed. For example, in the analysis of NAS-Bench-360 [4], a recent benchmark designed for improving task diversity in NAS evaluation, the authors showed a significant gap between models found by NAS methods, such as DARTS [2] and DenseNAS [5], and hand-crafted expert architectures on a number of distinct applications.

To improve the generalizability of AutoML methods, recent works such as AutoML-Zero [6] and XD-operations [7] propose to relax the inductive biases encoded in the standard search spaces. However, the former is not designed for practical deployment, and the latter is too expensive to execute even for simple problems like CIFAR-100 (Fig. 1b). Hence we ask: *is there an approach that can provide sufficient expressivity to yield high accuracy across multiple domains, while still retaining the faster search and the more efficient final models of discrete architecture search?*

---

[*] Equal contribution.

36th Conference on Neural Information Processing Systems (NeurIPS 2022).

In this work, we answer in the affirmative by introducing a novel NAS method called DASH (**D**iverse-task **A**rchitecture **S**earc**H**). In order to attain *multi-domain capability*, DASH adapts standard CNN backbones to various learning problems by finding substitutes for their layer operations. Concretely, we consider a search space of cross-scale dilated convolutions which are effective for multi-scale feature extraction [8, 9] and context aggregation [10, 11]. Our key difference from past search spaces is that we explicitly consider filters with *a wide range of kernel sizes and dilations*—while most NAS methods only handle kernels with maximum size 5 and dilation rate 2, our proposed operator space includes not only the conventional small kernels but also significantly larger ones with size 15 or dilation 127 (Fig. 2). This design choice is motivated by the fact that large kernels can

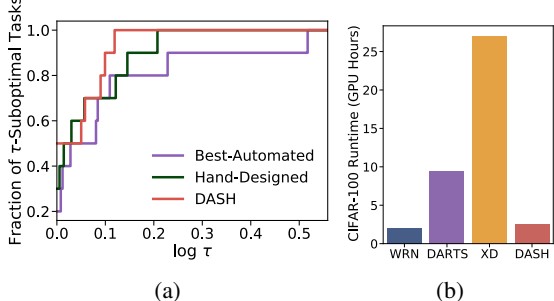

(a)                    (b)

Figure 1: (a) Comparing the aggregate performance of the best AutoML methods (task-wise), hand-designed models, and DASH on ten diverse tasks via performance profiles (defined in Section 4.1). Larger values (larger fractions of tasks on which a method is within $\log \tau$-factor of the best) are better. (b) Runtime for Wide ResNet, DARTS, XD, and DASH on CIFAR-100. XD is too expensive to be applied to other tasks considered in this work [4, 7].

capture input relations for dense prediction problems [12], model long-range dependencies for sequence tasks [13, 14], and resemble global-attention in Transformers [15]. Thus, DASH's cross-scale search space enables adaptation to diverse downstream tasks, unlike prior NAS work which targets image classification and assumes that small kernels are sufficient.

However, *efficiently* searching for an appropriate kernel configuration in this expansive cross-scale search space is non-trivial. Indeed, for existing NAS algorithms, the cost of exploring a combinatorially large set of operators is substantial. Even for weight-sharing methods that are known for efficiency, e.g., DARTS [16], the computational complexity scales directly with the number of kernels considered and quadratically with the largest kernel size. To overcome this obstacle, DASH explores multi-scale convolutions via three techniques—the first two exploit mathematical properties of convolutions, and the last one takes advantage of fast matrix multiplication on GPUs. Specifically:

1. Using the **linearity** of convolutions, we mix several convolutions by computing one convolution equipped with a combined kernel rather than applying each filter separately and aggregating multiple outputs. While the number of convolution computations required by the naive aggregation of $|K|$ possible kernel sizes and $|D|$ possible dilations is $\mathcal{O}(|K||D|)$, our approach has $\mathcal{O}(1)$ complexity, independent of the search space size.

2. Using the **diagonalization** of convolutions, we relegate a major portion of the computation to element-wise multiplication in the Fourier domain, minimizing the effect of the largest kernel size on the complexity of our algorithm. For instance, a standard 1D convolution requires $\mathcal{O}(nk)$ operations to convolve a size-$k$ kernel with a length-$n$ input, but a Fourier convolution takes only $\mathcal{O}(n \log n)$, a critical improvement that makes searching over large kernels significantly easier.

3. Our final strategy is to use Kronecker products of undilated kernels and small sparse matrices to compute dilated kernels quickly on GPUs. This brings an additional two-fold speedup on top of the previous techniques.

Aside from these innovations, DASH employs the standard weight-sharing scheme of training a supernet, discretizing to obtain a model, and retraining the model for end tasks [2]. We analyze the asymptotic complexity of the first two techniques and verify the practical utility when all three are combined together. In particular, DASH achieves a ten-fold speedup in total for differentiable NAS over the multi-scale search space. Moreover, we show that searching over large kernels is necessary to solve diverse problems and that each technique on its own cannot scale in this large-kernel setting.

In terms of accuracy performance, we evaluate DASH on ten datasets spanning multiple application areas such as PDE solving, protein folding, and disease prediction from NAS-Bench-360 [4]. As shown in Fig. 1, DASH yields models with better aggregate performance than those returned by leading AutoML methods as well as hand-designed task-specific architectures. As for individual tasks, DASH beats all past automated approaches on seven of the ten problems and exceeds hand-designed models on seven, simultaneously maintaining strong efficiency relative to weight-sharing methods

like DARTS. The empirical success of DASH implies that CNNs with appropriate kernels can be competitive baselines for problems where expert architectures are not available. Our code is made public at `https://github.com/sjunhongshen/DASH`.

## 2 Related Work

Neural architecture search (NAS) aims to automate the design of neural networks. Recently, there has been significant progress in both search space design [17, 2, 7] and search strategy development [1, 18, 2, 19, 3]. However, these methods are mostly evaluated on image classification or segmentation, with a few focusing on new applications such as image restoration [20], audio classification [21], and machine translation [22]. Problems beyond the vision and language domains are less-explored.

In this work, we seek to improve the generalizability of NAS by a morphism-based approach [23, 24, 7] that adapts existing CNN backbones to target tasks. We specifically focus on expanding the operation space to multiple types of convolutions, simultaneously varying the kernel size and the dilation factor. Past work in related directions has at most studied the easier problem of altering dilation alone, and only for vision tasks [25]. Therefore, although convolution has been an integral part of NAS, how to search over a large set of convolutional operators remains an open problem. In the following, we identify four types of solutions from existing work and illustrate their limitations.

**Differentiable Architecture Search (DARTS)** We can treat convolutions as ordinary operators and apply a scalable NAS algorithm. In particular, DARTS [2] introduces continuous relaxation to the weight-sharing paradigm [1] and allows us to gain information about many networks efficiently by training a combined supernet. The algorithm relaxes the discrete set of operations at each edge in a computational graph as a softmax so the search process is end-to-end differentiable and amenable to regular optimizers. After search, it discretizes the weights to output a valid architecture. The original DARTS search space contains only four convolutions with kernel sizes no larger than 5 and dilation rates no larger than 2. While this small search space might be enough for low-resolution image input, it is insufficient for diverse tasks such as high-dimensional time series problems [4]. Although one could add more convolutions one-by-one to the operator set to augment performance, this approach scales poorly, as reflected in the limited search spaces of similar methods like AMBER [26]. In fact, AMBER has to shift the kernel size up to achieve good performance on long-sequence genomic data.

**MergeNAS and RepVGG** An alternative way to explore the convolutional search space is to take advantage of the operator's linearity. That is, we can first mix the kernels and then apply convolution once, unlike DARTS which computes each convolution separately and outputs the aggregated result to the next layer. This kernel-mixing strategy, which we call *mixed-weights* and will formally define in Section 3.2, has been employed by MergeNAS [27] and RepVGG [28] to improve architecture search robustness and VGG inference speed, respectively. It works well for *a few small* kernels. However, we will show later that similar to DARTS, *mixed-weights* on its own is also insufficient for searching over *a diverse set of large kernels* which is crucial to solving a wide range of problems.

**Expressive Diagonalization (XD)** Apart from linearity, XD-operations [7] propose to utilize the convolutional theorem in architecture search. In particular, XD expresses the convolution acting on input $\mathbf{x}$ with filter $\mathbf{w}$ as $\mathbf{K}\operatorname{diag}(\mathbf{Lw})\mathbf{Mx}$, where $\mathbf{K}, \mathbf{L}, \mathbf{M}$ are appropriate discrete Fourier transforms, and constructs an expansive search space by replacing these transforms with searched kaleidoscope matrices [29]. Although this new search space includes all types of convolutions, the search process is unacceptably long even for simple benchmarking tasks such as CIFAR-100 (Fig. 1b), let alone the more complex set of diverse problems that we consider in this paper. In addition, the output architectures of XD are as inefficient as the supernet due to the absence of a discretization step.

**Single-Path NAS** Lastly, Single-Path NAS [30] defines a large filter and uses its subsets for smaller filters. This DARTS-based method compensates operator heterogeneity for efficiency during search. It does not handle search spaces with many large kernels and is not evaluated on diverse tasks.

Outside the field of AutoML, there is also emerging interest in designing general-purpose models such as Perceiver IO [31] and Frozen Pretrained Transformer [32]. However, these Transformer-based models do not adapt the network to the target tasks and are generally harder to train compared with CNNs. In Table 2, we evaluate Perceiver IO and show that its performance is not ideal. Recent works also propose to learn the kernel sizes for CNNs from the data [33], but they take an entirely different approach from NAS by multiplying the response of a convolution layer with a learnable Gaussian mask.

---

**Algorithm 1** DASH

---

**Input:** training data $Z$, loss function $l$, the set of kernel sizes $K$, the set of dilation rates $D$, and subsampling ratio $p$

Initialize the backbone and replace each **Conv** layer with the mixed operation $\mathbf{AggConv}_{K,D}$

**while** not converged **do**

    Subsample $p|Z|$ training points uniformly at random

    Compute forward pass using Equation 4

    Descend the architecture parameters $\alpha$ by $\nabla_\alpha l(\mathbf{w}, \alpha)$ and the model weights $\mathbf{w}$ by $\nabla_\mathbf{w} l(\mathbf{w}, \alpha)$

**end while**

Select $\arg\max_{k \in K, d \in D} \alpha_{k,d}$ for each **AggConv** layer

Tune retraining hyperparameters on a validation subset of the training data

Retrain the discretized model with all training data

---

## 3 Methods

Now, we describe the details of DASH (Algorithm 1). We first explain how DASH leverages existing networks to initialize the supernet and generate different models for diverse tasks. Then, we formally define the multi-scale convolution search space and propose a fast way to search this space using the three efficiency-motivated techniques mentioned earlier. Finally, we outline the procedure for discretizing the search space and retraining the searched model.

### 3.1 Decoupling Topology and Operations

Every architecture is a mapping from model weights to functions and can be described by a directed acyclic graph $G(V, E)$. Each edge in $E$ is characterized by $(u, v, \mathbf{Op})$, where $u, v \in V$ are nodes and $\mathbf{Op}$ is an operation applied to $u$. Node $v$ aggregates the outputs of its incoming edges. NAS aims to automatically select the edge operations and the graph topology to optimize some objective. For each edge, $\mathbf{Op}$ is chosen from a search space $S = \{\mathbf{Op}_a | a \in A\}$ where $a \in A$ are architecture parameters. In past work, $A$ usually indexes a small set of operations. For instance, the DARTS search space specifies $A_{discrete} = \{1, \ldots, 8\}$ with $S = \{\mathbf{Zero}, \mathbf{Id}, \mathbf{MaxPool}_{3\times3}, \mathbf{AvgPool}_{3\times3}, \mathbf{Conv}_{3\times3}, \mathbf{Conv}_{5\times5}, \mathbf{DilatedConv}_{3\times3,2}, \mathbf{DilatedConv}_{5\times5,2}\}$. However, $A$ can also be defined systematically to identify operator properties, e.g., $\{$(kernel size $k$, dilation $d)\}$ for convolutions.

A common way to determine the network topology is to search for blocks of operations and stack several blocks together. In this work, we take a different, morphism-based approach [23, 24, 7]: we use existing networks as backbones and replace certain layers in the backbone with the searched operations. Specifically, we select a set of architectures to accommodate both 2D and 1D datasets. Convolutional layers with different kernels can then be plugged into these networks. An advantage of decoupling topology and operation search is flexibility: the searched operators can vary from the beginning to the end of a network, so features at different granularities can be processed differently.

We pick Wide ResNets (WRNs) [34, 35] as the backbone networks due to their simplicity and effectiveness in image and sequence modeling. Before search, the supernet is initialized to the backbone. Then, all **Conv** layers are substituted with an operator $\mathbf{AggConv}_{K,D}$ (short for aggregated convolution) that represents the new search space which we now define. For simplicity, our mathematical discussion will stick to the 1D case, though our experiments are on both 1D and 2D data.

### 3.2 Efficiently Searching for Multi-Scale Convolutions

A convolution filter is specified by the kernel size $k$ and the dilation rate $d$ (we do not consider stride which does not change the filter shape). The effective filter size is $(k-1)d + 1$ with nonzero entries separated by $d - 1$ zeros. Let $\mathbf{Conv}_{k,d}$ be the convolution with kernel size $k$, dilation rate $d$, $c_{in}$ input channels, and $c_{out}$ output channels. Given input data with shape $n$, let $K$ be our interested set of kernel sizes, $D$ the set of dilations. We define the $\mathbf{AggConv}_{K,D}$ search space as

$$S_{\mathbf{AggConv}_{K,D}} = \{\mathbf{Conv}_{k,d} | k \in K, d \in D\}. \tag{1}$$

Table 1: Complexity of different methods for computing **AggConv**. For notation, $\bar{K} := |D| \cdot \sum_{k \in K} k$, $\bar{D} := \max_{k,d}(k-1)d + 1$. A detailed analysis is provided in Appendix A.2.

| Method | MULTs | ADDs |
|---|---|---|
| *mixed-results* (Eqn. 2) | $(c_{in}c_{out}\bar{K} + c_{out}|K||D|)n$ | $(c_{in}c_{out}\bar{K} + c_{out}|K||D|)n$ |
| *mixed-weights* (Eqn. 3) | $c_{in}c_{out}(\bar{K} + \bar{D}n)$ | $c_{in}c_{out}\bar{D}(|K||D| + n)$ |
| DASH (Eqn. 4) | $c_{in}c_{out}(\bar{K} + n) + \mathcal{O}(c_{in}c_{out}n\log n)$ | $c_{in}c_{out}(|K||D|\bar{D} + n) + \mathcal{O}(c_{in}c_{out}n\log n)$ |

Hence, $A = K \times D$ in previous notations. $S_{\mathbf{AggConv}_{K,D}}$ contains a collection of convolutions with receptive field size ranging from $k_{min}$ to $d_{\max}(k_{\max} - 1) + 1$, which allows us to discover models that process the input with varying resolution at each layer.

To retain the efficiency of discrete NAS, we apply the continuous relaxation scheme of DARTS to $S_{\mathbf{AggConv}_{K,D}}$, which mixes all operations in the space using architecture parameters $\{\alpha_{k,d} \in \triangle_{|K||D|} | (k,d) \in K \times D\}$[1], so the output of each edge in the computational graph is

$$\mathbf{AggConv}_{K,D}(\mathbf{x}) := \sum_{k \in K} \sum_{d \in D} \alpha_{k,d} \cdot \mathbf{Conv}(\mathbf{w}_{k,d})(\mathbf{x}). \tag{2}$$

Here $\{\mathbf{w}_{k,d} | (k,d) \in K \times D\}$ are the kernel weights. The resulting supernet can be trained end-to-end, and our hope is that after search, the most important operation is assigned the highest weight. However, the complexity of computing the above summation directly, a baseline algorithmic approach we call **mixed-results**, is $\mathcal{O}(c_{in}c_{out}(|K||D| + \bar{K})n)$, where $\bar{K} := |D| \sum_{k \in K} k$. *Mixed-results* can be expensive when we increase the maximum element in $K$ or $D$ with larger input size $n$. To improve upon it, we propose three techniques which build up to the efficiency-oriented DASH.

### 3.2.1 Technique 1: Mixed-Weights

Since convolution is linear, instead of computing $|K||D|$ convolutions, we can combine the kernels and compute convolution once. We call this approach **mixed-weights**:

$$\mathbf{AggConv}_{K,D}(\mathbf{x}) = \mathbf{Conv}\left(\sum_{k \in K} \sum_{d \in D} \alpha_{k,d} \cdot \mathbf{w}'_{k,d}\right)(\mathbf{x}). \tag{3}$$

Here $\mathbf{w}'$ is the properly padded version of $\mathbf{w}$ (appending 0's at the end of each dimension) so that filters of different sizes can be added. The aggregated kernel has size $\bar{D} := \max_{k,d}(k-1)d + 1$ and the $n$-dependent term of the complexity of *mixed-weights* is $c_{in}c_{out}\bar{D}n$. Hence, it removes the direct dependence of the leading-order term on $|K||D|$, the search space size, that *mixed-results* had.

### 3.2.2 Technique 2: Fourier Convolution

If we wish to increase the kernel size and dilation with the input size, the complexity of *mixed-weights* will still grow *implicitly* with the search space size through the dependence on $\bar{D}$. To address this issue, we combine the kernel re-weighting idea with another technique motivated by the convolution theorem. Given a kernel $\mathbf{w}$, recall that $\mathbf{Conv}(\mathbf{w})(\mathbf{x}) = \mathbf{F}^{-1} \operatorname{diag}(\mathbf{Fw}')\mathbf{Fx}$, where $\mathbf{F}$ is the discrete Fourier transform (DFT) and $\operatorname{diag}(\mathbf{z})$ is a diagonal matrix with entries $\mathbf{z}$. In other words, convolution involves multiplying the DFT of the kernel by that of the input. Since the DFT can be applied in time $\mathcal{O}(n \log n)$ using the Fast Fourier Transform (FFT) and apart from that we only need element-wise multiplication, this yields an efficient approach to reducing dependence on the combined kernel size $\bar{D}$. While Roberts et al. [7] replaced the DFTs with a *continuous* set of matrices for XD-operations, our approach can be viewed as replacing the middle DFT with a *discrete* set of matrices for the purpose of efficiency. Accordingly, DASH computes $\mathbf{AggConv}_{K,D}$ as follows:

$$\mathbf{AggConv}_{K,D}(\mathbf{x}) = \mathbf{F}^{-1} \operatorname{diag}\left(\mathbf{F}\left(\sum_{k \in K} \sum_{d \in D} \alpha_{k,d} \cdot \mathbf{w}'_{k,d}\right)\right)\mathbf{Fx}. \tag{4}$$

Note that while the kernel changes for each $\mathbf{Conv}_{k,d}$, the input does not. Hence, we also save time by transforming the input to the frequency domain only once.

---

[1]$\triangle$ denotes the probability simplex.

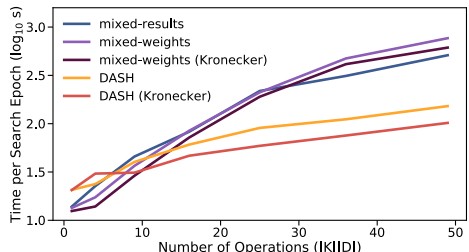
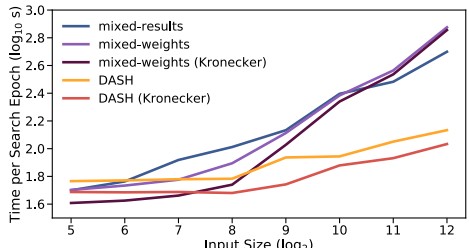

Figure 2: $\log_{10}$ time for one search epoch vs. number of ops in $S_{\mathbf{AggConv}_{K,D}}$. We vary the search space by letting $K = \{2p + 1 | 1 \leq p \leq c\}$, $D = \{2^q - 1 | 1 \leq q \leq c\}$ and increasing $c$ from 1 to 7.

Figure 3: $\log_{10}$ time for one search epoch vs. input length of single-channel 1D data. We fix $K = \{3, 5, 7, 9, 11\}$, $D = \{1, 3, 7, 15, 31\}$ and test $n \in \{2^5, \ldots, 2^{12}\}$.

In Table 1, we report the theoretical complexities of the baselines and DASH, the latter leveraging both Technique 1 and 2. It is easy to obtain the operation complexities of *mixed-weights* and *mixed-results*. For DASH, the number of multiplications and additions can be attributed to the inner weight sum and multi-channel product (the first term) as well as three FFTs (the second). A detailed analysis is provided in Appendix A.2. We see that *mixed-weights* is favorable to *mixed-results* when $\bar{D} < \bar{K} = \mathcal{O}(|D|k_{\max}^2)$, which occurs with large kernels and a few dilations. On the other hand, only DASH completely separates the dominant terms containing $c_{in}c_{out}n$ from the size of the search space and its elements, replacing them by $\mathcal{O}(\log n)$, which is small for any realistic $n$. As we increase $k_{\max}$ and $d_{\max}$ for larger inputs, this will also lead to a slower asymptotic increase in complexity, making DASH an attractive choice for the multi-scale search space where $\bar{D}$ is large by design to extract possible long-range dependencies in the data.

### 3.2.3 Technique 3: Kronecker Dilation

To efficiently implement the kernel summation in *mixed-weights* and DASH on a GPU, we introduce our final technique: after initializing $\mathbf{w}_{k,d}$ for each $\mathbf{Conv}_{k,d}$ separately, we use a Kronecker product $\otimes$ to transform the undilated kernels into dilated forms. For example, to compute a 2D convolution with dilation $d$, we introduce the sparse pattern matrix $\mathbf{P} \in \mathbb{R}^{d \times d}$ whose entries are all 0's except for the upper-left entry $\mathbf{P}_{1,1} = 1$:

$$\mathbf{P} = \begin{bmatrix} 1 & 0 & \cdots & 0 \\ 0 & 0 & \cdots & 0 \\ \vdots & \vdots & \ddots & \vdots \\ 0 & 0 & \cdots & 0 \end{bmatrix}. \tag{5}$$

Then, $\mathbf{w}_{k,d} = \mathbf{w}_{k,1} \otimes \mathbf{P}$. Beyond the theoretical gains shown in Wu et al. [36], this dilation strategy is empirically faster than the standard way of padding 0's into $\mathbf{w}_{k,1}$ (Fig. 2 and 3). After dilating the kernels, we sum them together, zero-pad to match the input size, and apply the FFTs.

### 3.2.4 Ablation Study for the Proposed Techniques

To check that our asymptotic analysis leads to actual speedups and perform an ablation study on the proposed techniques, we evaluate the three methods on single-channel 1D input (experiment details are in Appendix A.3). Since both *mixed-weights* and DASH require kernel summation, which can be implemented with Kronecker dilation (Technique 3), we compare five methods in total.

Fig. 2 illustrates the combined forward- and backward-pass time in log scale for one search epoch vs. the size of $S_{\mathbf{AggConv}_{K,D}}$ when $n = 1000$. For small $\bar{D}$, the FFT overhead makes DASH runtime slightly longer but the difference is negligible. However, as $\bar{D}$ increases, the DASH curves grow much slower whereas the runtimes for the other methods scale with the number of operations. In Fig. 3, we fix $K$ and $D$ to study how runtime is affected by input size $n$. Both *mixed-results* and *mixed-weights* become extremely inefficient for large $n$'s which commonly occur in time-series or signal processing. Surprisingly, DASH's runtime does not increase much with $n$. We hypothesize that this is due to wallclock-time being dominated by data-passing at that speed.

In general, Technique 1 on its own scales poorly for the considered search space. This is why methods like MergeNAS [27] cannot be used in our setting. Though XD makes use of Technique 2, it considers a parametrized space with infinitely many operations that need to be continuously evolved and is too expensive to be applied to tasks beyond CIFAR-100 (c.f. Fig. 1b). Technique 3 contributes to $2\times$ speedups for both *mixed-weights* and DASH. Overall, DASH (Kronecker) leads to about $10\times$ search-time speedups compared to the *mixed-results* scheme of DARTS for both the large operation space and large input size regimes. Hence, we use this version of DASH in later experiments.

### 3.3 The Full Pipeline: Architecture Search, Hyperparameter Optimization, and Retraining

Having shown the main techniques for searching a large space of kernel patterns, we now detail the full search and model development pipeline. Given a dataset, we set $K = \{3 + d(p-1)|1 \le p \le p_{\max}\}$ for kernel sizes and $D = \{2^q - 1|1 \le q \le q_{\max}\}$ for dilations. For 2D input, we set $d$ to 2, $p_{\max}$ to 4, and $q_{\max}$ to 4. For longer 1D sequence data, we set $d = 4$, $p_{\max} = 5$, and $q_{\max} = 4$. For instance, CIFAR has size $3 \times 32 \times 32$ where 3 is the number of channels and $n = 32$. The corresponding $K$ is $\{3, 5, 7, 9\}$ and $D$ is $\{1, 3, 7, 15\}$. To normalize architecture parameters into a probability distribution, we adopt the soft Gumbel Softmax activation, similar to Xie et al. [18].

The backbone networks are as follows. For 2D tasks, we use WRN 16-1 as the search backbone to accelerate supernet training and WRN 16-4 for retraining. For 1D tasks, we use 1D WRN [35] in the entire pipeline. During search, we subsample the training data at each epoch. Given the loss for the target task, DASH jointly optimizes the model weights and the architecture parameters using direct gradient descent. This single-level optimization is more efficient than two-stage NAS, which finds initial assignments for architecture parameters and trains the candidates from scratch.

After searching for a predefined number of epochs, we discretize the search space and pick $\mathbf{Conv}_{k,d} \in S_{\mathbf{AggConv}_{K,D}}$ with the largest weight for each layer. The final model has a similar overall structure to the backbone, but the intermediate operations are tailored to the target task. To improve training stability, we additionally add a simple hyperparameter tuning stage between search and retraining using grid search (configuration space shown in Appendix A.5.2).

For each setting, we train the discretized model on *a subset* of the training data for *fewer* epochs so the tuning cost is a small fraction of the entire pipeline's cost (Table 6), Then, we evaluate the performance on a holdout validation set and select the configuration with the best validation score. As a final step, we retrain the discretized model with the optimal hyperparameters on all training data until convergence. Like other weight-sharing methods with discretization, our final model will be more efficient than the supernet.

## 4 Evaluation

We evaluate the performance of DASH on diverse tasks using ten datasets from NAS-Bench-360 [4], a benchmark spanning multiple application domains, input dimensions, and learning objectives.[2] These include classical vision tasks such as CIFAR-100 where CNNs do well, scientific computing tasks such as Darcy Flow where standard CNN backbones can perform poorly [7, 37], sequence tasks such as DeepSEA where large dilations are preferred [13, 26], and many others. Thus, our evaluation will not only test whether DASH can find good architectures in the proposed new search space, but also investigate whether multi-scale convolution is a strong competitor for solving different problems. In fact, our results show that DASH is a top choice for many tasks, obtaining in-aggregate the best speed-accuracy trade-offs among the methods we evaluate (c.f. Fig. 5).

### 4.1 Baselines and Experimental Setup

For each NAS-Bench-360 task, we compare DASH with the following methods: DenseNAS [5] and GAEA PC-DARTS [3], which represent general NAS; Auto-DeepLab [38] and AMBER [26], which represent specialist NAS methods for dense prediction and 1D tasks, respectively; 1D temporal convolutional network (TCN) [13], regular WRN, and WRN with hyperparameter tuner ASHA [39], which represent natural NAS baselines; and Perceiver IO [31], which represents non-NAS general-purpose models. While these results are available in Tu et al. [4], we additionally add a BABY DASH

---

[2]For completeness, we give a task summary in the Appendix.

Table 2: Error rates (lower is better) on NAS-Bench-360 tasks across diverse application domains and problem dimensions (the last three problems are 1D and the rest are 2D). DASH beats *all the other NAS methods* on 7/10 tasks and exceeds hand-designed expert models on 7/10 tasks. Scores of DASH are averaged over three trials. Scores of the baselines are from Tu et al. [4]. See Table 5 in the Appendix A.6 for standard deviations.

| | CIFAR-100 | Spherical | Darcy Flow | PSICOV | Cosmic | NinaPro | FSD50K | ECG | Satellite | DeepSEA |
| --- | --- | --- | --- | --- | --- | --- | --- | --- | --- | --- |
| | 0-1 error (%) | 0-1 error (%) | relative $\ell_2$ | $MAE_8$ | 1-AUROC | 0-1 error (%) | 1-mAP | 1-F1 | 0-1 error (%) | 1- AUROC |
| Expert | **19.39** | 67.41 | 0.008 | 3.35 | **0.13** | 8.73 | 0.62 | **0.28** | 19.8 | 0.30 |
| WRN | 23.35 | 85.77 | 0.073 | 3.84 | 0.24 | 6.78 | 0.92 | 0.43 | 15.49 | 0.40 |
| TCN | - | - | - | - | - | - | - | 0.57 | 16.21 | 0.44 |
| WRN-ASHA | 23.39 | 75.46 | 0.066 | 3.84 | 0.25 | 7.34 | 0.91 | 0.43 | 15.84 | 0.41 |
| DARTS-GAEA | 24.02 | **48.23** | 0.026 | **2.94** | 0.22 | 17.67 | 0.94 | 0.34 | 12.51 | 0.36 |
| DenseNAS | 25.98 | 72.99 | 0.10 | 3.84 | 0.38 | 10.17 | 0.64 | 0.40 | 13.81 | 0.40 |
| Auto-DL | - | - | 0.049 | 6.73 | 0.49 | - | - | - | - | - |
| AMBER | - | - | - | - | - | - | - | 0.67 | 12.97 | 0.68 |
| Perceiver IO | 70.04 | 82.57 | 0.24 | 8.06 | 0.48 | 22.22 | 0.72 | 0.66 | 15.93 | 0.38 |
| BABY DASH | 25.56 | 63.45 | 0.016 | 3.94 | 0.16 | 8.28 | 0.62 | 0.37 | 13.29 | 0.37 |
| DASH | 24.37 | 71.28 | **0.0079** | 3.30 | 0.19 | **6.60** | **0.60** | 0.32 | **12.28** | **0.28** |

baseline: we run DASH in the DARTS convolution space with $K = \{3, 5\}$ and $D = \{1, 2\}$ to study whether large kernel sizes and dilations are necessary to strong performance across-the-board. Finally, we compare our method to the expert architectures selected by NAS-Bench-360. These models are representatives of the best that hand-crafting has to offer.

Each dataset is preprocessed and split using the NAS-Bench-360 script, with the training set being used for search, hyperparameter tuning, and retraining. To construct the multi-scale search space, we set $K$ and $D$ according to the rules in Section 3.3. We use the default SGD optimizer for the WRN backbone and fix the learning rate schedule as well as the gradient clipping threshold for every task. The entire DASH pipeline can be run on a single NVIDIA V100 GPU, which is also the system that we use to report the runtime cost. Full experimental details can be found in the Appendix.

We evaluate the performance of all competing methods following the NAS-Bench-360 protocol. We first report results of the target metric for each task by running the model of the *last* epoch on the test data. Then, we report aggregate results via *performance profiles* [40], a technique that considers both outliers and small performance differences to compare methods across multiple tasks robustly. In such plots, each curve represents one method. The $\tau$ on the $x$-axis denotes the fraction of tasks on which a method is no worse than a $\tau$-factor from the best.

### 4.2 Results and Discussion

We present the accuracy results for each task in Table 2 and the performance profiles in Fig. 4. Fig. 4 clearly demonstrates that DASH is superior to other competing methods in terms of aggregate performance. In particular, it ranks first among all *automated* models for 7/10 tasks, among all *expert* models for 7/10 tasks, and performs favorably when considering both accuracy and efficiency as shown in Fig. 5. In addition, Table 3 shows that DASH outperforms DARTS in speed for all 10 tasks (in several cases by an order of magnitude), and attains comparable efficiency with training vanilla WRNs for 6/10 tasks (full-pipeline time is less than or about twice as long as the WRN training time). In the following, we provide a detailed analysis of the experimental results.

**DASH dominates automated methods.** Compared to other automated methods, DASH has a clear advantage in accuracy. Even for tasks where it does not beat the expert, e.g. ECG, DASH's performance is significantly better than other AutoML methods. It also outperforms specialist methods Auto-DL and AMBER on dense prediction and 1D tasks, respectively. Although DARTS does best on CIFAR-100 (the task for which it was designed), Spherical, and PSICOV, it is the worst on NinaPro and FSD50K. Note that the underperformance of DASH on CIFAR-100 relative to WRN (and on Spherical and Cosmic relative to BABY DASH) suggests suboptimality of the gradient descent optimization procedure but not of the operation space, since WRN and BABY DASH are contained in our search space. This indicates a future direction to improve optimization in the DASH search space.

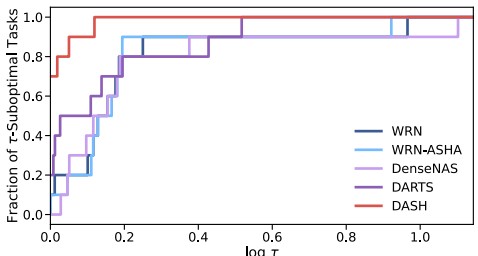
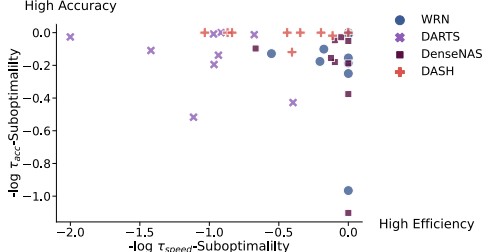

Figure 4: Performance profiles of general NAS methods and DASH on NAS-Bench-360. DASH being far in the top left corner indicates it is rarely suboptimal and is often the best.

Figure 5: Comparing $-\log \tau$-suboptimality of speed vs. accuracy on all tasks. DASH's concentration in the top right corner indicates its strong efficacy-efficiency trade-offs relative to the other methods.

**DASH dominates expert architectures.** While the degree of sophistication of the expert networks varies task by task, the performance of DASH on tasks such as Darcy Flow suggests that it is capable of competing with highly specialized networks, e.g., Fourier Neural Operator [37] for PDE solving. These results imply that DASH, and more generally the strategy of equipping backbone networks with task-specific kernels, is a promising approach for tackling model development in new domains. Meanwhile, DASH consistently outperforms Perceiver IO which represents non-automated general-purpose models. We speculate that the poor performance of Perceiver IO is because it is developed on more sophisticated natural language and multi-modal reasoning tasks and training Transformers is generally difficult.

**Large kernels are needed.** We also ablate the large-$k$-large-$d$ design of the search space by comparing DASH with BABY DASH. We hypothesize that for the same task, a small performance gap between the two methods would indicate that small kernels suffice for extracting local features, whereas a major degradation in the quality of the BABY DASH model can imply that the task needs global modeling. Consequently, datasets such as Darcy Flow and ECG provide compelling evidence that kernels with large receptive fields play an important role in solving real-life problems and further back up the design of our multi-scale convolutional search space. An example of the series of WRN kernels found by DASH on Darcy Flow is: $\mathbf{Conv}_{5,3} \rightarrow \mathbf{Conv}_{3,1} \rightarrow \mathbf{Conv}_{3,1} \rightarrow \mathbf{Conv}_{3,15} \rightarrow \mathbf{Conv}_{7,15} \rightarrow \mathbf{Conv}_{9,7} \rightarrow \mathbf{Conv}_{9,7} \rightarrow \mathbf{Conv}_{3,7} \rightarrow \mathbf{Conv}_{5,7} \rightarrow \mathbf{Conv}_{5,15} \rightarrow \mathbf{Conv}_{9,7} \rightarrow \mathbf{Conv}_{3,7} \rightarrow \mathbf{Conv}_{7,7}$. We can see that large kernels are indeed selected during search. More visualizations can be found in Appendix A.8.

Table 3: Full-pipeline runtime in GPU hours for NAS-Bench-360 (PSICOV results are omitted due to a discrepancy in the implementation of data loading). DASH is consistently faster than DARTS, and it is less than a factor of two slower than simply training a WRN on six of the ten tasks. DenseNAS is fast but its accuracy is far less impressive. XD is too expensive to be applied to tasks other than CIFAR-100 [7].

| Task | DARTS | DenseNAS | WRN | DASH |
|---|---|---|---|---|
| CIFAR-100 | 9.5 | 2.5 | 2 | 2.5 |
| Spherical | 16.5 | 2.5 | 2 | 5 |
| Darcy Flow | 6.5 | 0.5 | 0.5 | 5.3 |
| Cosmic | 21.5 | 2.5 | 4 | 6.8 |
| NinaPro | 0.5 | 0.2 | 0.2 | 0.3 |
| FSD50K | 37 | 4.5 | 4 | 29 |
| ECG | 140 | 6.5 | 5 | 1.3 |
| Satellite | 28 | 3 | 4.5 | 6.5 |
| DeepSEA | 39.5 | 2 | 1.5 | 10 |

**DASH is computationally efficient.** In addition to prediction quality, we also care about the efficiency of model selection. Table 3 provides the combined search and retraining time in GPU hours for DARTS, DenseNAS, and DASH, as well as the training time for vanilla WRN 16-4 without hyperparameter tuning (baseline results are taken from Tu et al. [4]). We also present the breakdown of DASH's full-pipeline runtime in Appendix A.7. A key observation is that the cost of DASH is consistently below DARTS' on all tasks and is similar to training a simple CNN for more than half of them. Although DenseNAS stands out by speed, its practical performance is less impressive.

In Fig. 5, we visualize the trade-off between efficiency and effectiveness for each method and task combination. Evidently, DASH takes an important step towards bridging the gap between the efficiency of DARTS and the expressivity of XD in NAS. The fact that DASH can be trained at a

low cost testifies that we need not sacrifice efficiency for adding more operations. In fact, we have actually shown that DASH is *both* faster *and* more effective than DARTS in many tasks.

**DASH works with other backbones and is backward compatible.** Lastly, in addition to the Wide ResNet backbone and NAS-Bench-360 tasks, we have also verified the efficacy of DASH on other backbones including TCN [13] and ConvNeXt [15], and on large-scale datasets including ImageNet in Appendix A.9. In particular, DASH is able to achieve a 1.4% increase in top-1 accuracy for ImageNet on top of the ConvNeXt backbone. As the latter was itself developed in part via manual tuning of the kernel size, this means that DASH outperforms human hand-tuning on ImageNet. These results show that DASH is backbone-agnostic and also works well with computer vision tasks, making it backward compatible with the original use cases of CNNs.

### 4.3 Limitations and Future Work

There are several open problems which we leave for future work. First, it is beneficial to study why certain kernel patterns are chosen, as the selected operations can hint us at the intrinsic properties of the datasets. Second, one can improve upon DASH, e.g., by including non-square convolutions for 2D problems, using a better optimization algorithm, or developing techniques that further reduces the memory usage of performing a forward architecture search pass. One could also construct a more comprehensive search space with high-level operators such as self-attention [41].

Meanwhile, although this paper focuses on NAS, which is an alternative to fine-tuning pretrained models, the aggregated convolution can be a plug-and-play module for algorithms that search for large-scale models. For instance, many Transformer models still depend on convolutions for feature extraction and transformation, and their performance relies on the quality of the embedded features. Since DASH is applicable to any architecture with a convolutional layer, it can be helpful for such models, including Vision Transformer with a convolutional patching layer [42], Deformable Transformer with a ResNet embedder [43], Swin Transformer with a convolutional decoder [44], and many others.

**Societal Impact** This paper calls for ML community's attention to less-studied application domains and moves towards truly democratizing machine learning in real life. In terms of broader societal impact, our work can exert a positive influence as it contributes to NAS efficiency and reduces the computational burden on AutoML end-users. However, lowering the barrier for applying ML to a wide range of tasks necessarily comes with the risk of misuse. Hence, it is imperative to develop NAS methods with privacy, safety, and fairness guarantees.

## 5 Conclusion

In this paper, we argue that a crucial goal of NAS is to discover accessible models for diverse tasks. To this end, we propose DASH, which efficiently searches for convolution patterns and integrates them into existing backbones. DASH overcomes the computational limitations of differentiable NAS and obtains high-quality models with accuracy comparable to or better than that of the handcrafted networks on many tasks. Our experiments show that convolution can be a universal operator for many under-explored areas. DASH is also a promising step towards developing general-purpose models with more complicated structures.

## Acknowledgments

We thank Maria-Florina Balcan, Nicholas Roberts, and Renbo Tu for providing useful feedback. This work was supported in part by DARPA FA875017C0141, the National Science Foundation grants IIS1705121, IIS1838017, IIS2046613 and IIS-2112471, an Amazon Web Services Award, a Facebook Faculty Research Award, funding from Booz Allen Hamilton Inc., a Block Center Grant, and a Facebook Fellowship Award. Any opinions, findings and conclusions or recommendations expressed in this material are those of the author(s) and do not necessarily reflect the views of any of these funding agencies.

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
