# A Appendix

## A.1 Term Clarification

Since we compare with a variety of methods in the paper, here we clarify some of the terms we use.

| what we say | what we are referring to |
|---|---|
| Best-Automated (Fig. 1a) | WRN, WRN-ASHA, DARTS, DenseNAS, Auto-DL, AMBER |
| Hand-Designed (Fig. 1a) | Expert architectures in Table 4 |
| AutoML | WRN-ASHA, DARTS, DenseNAS, Auto-DL, AMBER |
| NAS | DARTS, DenseNAS, Auto-DL, AMBER |
| WRN | WRN without hyperparameter tuning |

## A.2 Asymptotic Analysis

In this section we outline the runtime analysis used to populate the asymptotic complexities in Table 1. All three methods in the table—*mixed-results*, *mixed-weights*, and DASH—are computing the following weighted sum of convolutions:

$$\mathbf{AggConv}_{K,D}(\mathbf{x}) := \sum_{k \in K} \sum_{d \in D} \alpha_{k,d} \cdot \mathbf{Conv}(\mathbf{w}_{k,d})(\mathbf{x}). \tag{6}$$

We consider 1D inputs $\mathbf{x}$ with length $n$ and $c_{in}$ input channels; the convolutions have $c_{out}$ output channels. We view $\mathbf{Conv}(\mathbf{w}_{k,d})(\mathbf{x})$ as having the naive complexity $c_{in}c_{out}kn$ since the deep learning frameworks use the direct (non-Fourier) algorithm. *mixed-results* computes the sum directly, which involves (1) applying one convolution of each size $k$ and dilation to $\mathbf{x}$ at a cost of $c_{in}c_{out}kn$ MULTs and ADDs each for a total cost of $c_{in}c_{out}\bar{K}n$, (2) scalar-multiplying the outputs at a cost of $c_{out}|K||D|n$ MULTs, and (3) summing the results together at a cost of $c_{out}|K||D|n$ ADDs. *mixed-weights* instead (1) multiplies all kernels by their corresponding weight at a cost of $c_{in}c_{out}\bar{K}$ MULTs, (2) zero-pads the results to the largest effective kernel size $\bar{D}$ and adds them together at a cost of $c_{in}c_{out}|K||D|\bar{D}$ ADDs, and (3) applies the resulting $\bar{D}$-size convolution to the input at a cost of $c_{in}c_{out}\bar{D}n$ MULTs and ADDs. Finally, DASH also (1) does the first two steps of *mixed-weights* at a cost of $c_{in}c_{out}\bar{K}$ MULTs and $c_{in}c_{out}|K||D|\bar{D}$ ADDs but then (2) pads the resulting $\bar{D}$-size convolution to size $n$ and applies an FFT at a cost of $\mathcal{O}(c_{in}c_{out}n \log n)$ MULTs and ADDs, (3) applies an FFT to $\mathbf{x}$ at a cost of $\mathcal{O}(c_{in}n \log n)$, (4) element-wise multiplies the transformed filters by the inputs at a cost of $c_{in}c_{out}n$ MULTs, (5) adds up $c_{in}$ results for each of $c_{out}$ output channels at a cost of $c_{in}c_{out}$ MULTs, and (6) applies an iFFT to the result at a cost of $\mathcal{O}(c_{out}n \log n)$.

## A.3 Experiment Details for Fig. 2 and Fig. 3

For the speed tests, we work with the Sequential MNIST dataset, i.e., the 2D $28 \times 28$ images are stretched into 1D with length 784. We zero pad or truncate the input to generate data with different input size $n$. The backbone is 1D WRN with the same structure as introduced in Section 3. The batch size is 128. We run the workflow on a single NVIDIA V100 GPU. The timing results reported are the $\log_{10}$(combined forward and backward pass time for one search epoch).

In Fig. 2, we study how the size of our multi-scale convolution search space affects the runtimes of *mixed-results*, *mixed-weights*, and DASH for $n = 1000$ (zero-padded MNIST). We define $K = \{3 + 2(p - 1) | 1 \le p \le c\}$, $D = \{2^q - 1 | 1 \le q \le c\}$ and varies $c$ from 1 to 7. Consequently, the number of operations included in the search space grows from 1 to 49.

In Fig. 3, we study how the input size affects the runtimes of the three methods. We fix $K = \{3, 5, 7, 9, 11\}$, $D = \{1, 3, 7, 15, 31\}$ and vary $n$ from $2^5$ to $2^{12}$.

## A.4 Information About Tasks in NAS-Bench-360

Table 4: Information about evaluation tasks in NAS-Bench-360 [4].

| Task name | # Data | Data dim. | Type | License | Learning objective | Expert arch. |
|-----------|--------|-----------|------|---------|--------------------|--------------|
| CIFAR-100 | 60K | 2D | Point | CC BY 4.0 | Classify natural images into 100 classes | DenseNet-BC [45] |
| Spherical | 60K | 2D | Point | CC BY-SA | Classify spherically projected images into 100 classes | S2CN [46] |
| NinaPro | 3956 | 2D | Point | CC BY-ND | Classify sEMG signals into 18 classes corresponding to hand gestures | Attention Model [47] |
| FSD50K | 51K | 2D | Point (multi-label) | CC BY 4.0 | Classify sound events in log-mel spectrograms with 200 labels | VGG [48] |
| Darcy Flow | 1100 | 2D | Dense | MIT | Predict the final state of a fluid from its initial conditions | FNO [37] |
| PSICOV | 3606 | 2D | Dense | GPL | Predict pairwise distances between residuals from 2D protein sequence features | DEEPCON [49] |
| Cosmic | 5250 | 2D | Dense | Open License | Predict propablistic maps to identify cosmic rays in telescope images | deepCR-mask [50] |
| ECG | 330K | 1D | Point | ODC-BY 1.0 | Detect atrial cardiac disease from a ECG recording (4 classes) | ResNet-1D [51] |
| Satellite | 1M | 1D | Point | GPL 3.0 | Classify satellite image pixels' time series into 24 land cover types | ROCKET [52] |
| DeepSEA | 250K | 1D | Point (multi-label) | CC BY 4.0 | Predict chromatin states and binding states of RNA sequences (36 classes) | DeepSEA [53] |

## A.5 Evaluation of DASH on NAS-Bench-360

### A.5.1 Backbone Network Structure

**2D Tasks** We use the Wide ResNet 16-4 [34] as the backbone for all 2D tasks. The original model is made up of $16\ 3 \times 3$ conv followed by 6 WRN blocks with the following structure ($i \in \{1, 2, 3, 4, 5, 6\}$ indicates the block index):

| BatchNorm, ReLU | |
|---|---|
| Conv 1 | $16 \times 4 \times ((i+1)//2)$ $(k = 3, d = 1)$ filters, ReLU |
| Dropout | dropout rate $p$ |
| BatchNorm, ReLU | |
| Conv 2 | $16 \times 4 \times ((i+1)//2)$ $(k = 3, d = 1)$ filters , stride $= (i+1)//2$, ReLU |
| Add residual (apply point-wise conv first if $c_{in} \neq c_{out}$) | |

The output block consists of a BatchNorm layer, a ReLU activation, a linear layer, and a final activation layer which we modify according to the task learning objective, e.g., log softmax for classification and sigmoid for dense prediction. We set $p = 0$ in search and tune $p$ as a hyperparameter for retraining. We use the WRN code provided here: https://github.com/meliketoy/wide-resnet.pytorch.

**1D Tasks** We use the 1D WRN [35] as the backbone for all 1D tasks. The model is made up of 3 residual blocks with the following structure:

| Conv 1 | $c_{out}$ $(k = 8, d = 1)$ filters |
|--------|-----------------------------------|
| Dropout | dropout rate $p$ |
| BatchNorm, ReLU | |
| Conv 2 | $c_{out}$ $(k = 5, d = 1)$ filters |
| Dropout | dropout rate $p$ |
| BatchNorm, ReLU | |
| Conv 3 | $c_{out}$ $(k = 3, d = 1)$ filters |
| Dropout | dropout rate $p$ |
| BatchNorm, ReLU | |

In the original architecture, $c_{out} = 64$. We set $c_{out}$ to $\min(4^{\text{num\_classes}//10+1}, 64)$ to account for simpler tasks with fewer class labels. The output block consists of a linear layer and a activation layer which we modify according to the task learning objective, e.g., log softmax for classification and sigmoid for dense prediction. We set $p = 0$ in search and tune $p$ as a hyperparameter for retraining. We use the 1D WRN code provided here: `https://github.com/okrasolar/pytorch-timeseries`.

### A.5.2 DASH Pipeline Hyperparameters

**Search**

- Epoch: 100
- Optimizer: `SGD(momentum=0.9, nesterov=True, weight_decay=5e-4)` for both model weights and architecture parameters
- Model weight learning rate: 0.1 for point prediction tasks, 0.01 for dense tasks
- Architecture parameter learning rate: 0.05 for point prediction tasks, 0.005 for dense tasks
- Learning rate scheduling: decay by 0.2 at epoch 60
- Gradient clipping threshold: 1
- Softmax temperature: 1
- Subsampling ratio: 0.2

To constrain the size of the searched model, we can add a regularization term to the gradients of the architecture parameters of large kernels. We set the penalty to 1e-5 times the receptive field size.

**Hyperparameter tuning**

- Epoch: 80
- Configuration space:
    - Learning rate: $\{$1e-1, 1e-2, 1e-3$\}$
    - Weight decay: $\{$5e-4, 5e-6$\}$
    - Momentum: $\{0.9, 0.99\}$
    - Dropout rate: $\{0, 0.05\}$

**Retraining**

- Epoch: 200
- Learning rate scheduling: for 2D tasks, decay by 0.2 at epoch 60, 120, 160; for 1D tasks, decay by 0.2 at epoch 30, 60, 90, 120, 160

**Task-Specific Hyperparameters**

| 2D tasks | CIFAR-100 | Spherical | Darcy Flow | PSICOV | Cosmic | NinaPro | FSD50K |
|---|---|---|---|---|---|---|---|
| Batch size | 64 | 64 | 10 | 8 | 4 | 128 | 128 |
| Input size | (32, 32) | (60, 60) | (85, 85) | (128, 128) | (128, 128) | (16, 52) | (96, 101) |
| Kernel sizes $(K)$ | $\{3,5,7,9\}$ | $\{3,5,7,9\}$ | $\{3,5,7,9\}$ | $\{3,5,7,9\}$ | $\{3,5,7,9\}$ | $\{3,5,7,9\}$ | $\{3,5,7,9\}$ |
| Dilations $(D)$ | $\{1,3,7,15\}$ | $\{1,3,7,15\}$ | $\{1,3,7,15\}$ | $\{1,3,7,15\}$ | $\{1,3,7,15\}$ | $\{1,3,7,15\}$ | $\{1,3,7,15\}$ |
| Loss $(l)$ | Cross Entropy | Cross Entropy | L2 | MSE | BCE w. Logits | Focal | BCE w. Logits |

| 1D tasks | Satellite | ECG | DeepSEA |
|---|---|---|---|
| Batch size | 256 | 1024 | 256 |
| Input size | 46 | 1000 | 1000 |
| Kernel sizes $(K)$ | $\{3,7,11,15,19\}$ | $\{3,7,11,15,19\}$ | $\{3,7,11,15,19\}$ |
| Dilations $(D)$ | $\{1,3,7,15\}$ | $\{1,3,7,15\}$ | $\{1,3,7,15\}$ |
| Loss $(l)$ | Cross Entropy | Cross Entropy | BCE w. Logits |

## A.6 Accuracy Results on NAS-Bench-360 with Error Bars

Table 5: Error rates (lower is better) of DASH and the baselines on tasks in NAS-Bench-360. Methods are grouped into three classes: non-automated, automated, and the DASH family. Results of DASH are averaged over three trials using the models obtained after the last retraining epoch.

| | CIFAR-100 0-1 error(%) | Spherical 0-1 error(%) | Darcy Flow relative $\ell_2$ | PSICOV MAE$_8$ | Cosmic 1-AUROC |
|---|---|---|---|---|---|
| WRN | 23.35±0.05 | 85.77±0.71 | 0.073±0.001 | 3.84±0.053 | 0.24±0.015 |
| Expert | **19.39±0.20** | 67.41±0.76 | 0.008±0.001 | 3.35±0.14 | **0.13±0.01** |
| Perceiver IO | 70.04±0.44 | 82.57±0.19 | 0.24±0.01 | 8.06±0.06 | 0.48±0.01 |
| WRN-ASHA | 23.39±0.01 | 75.46±0.40 | 0.066±0.00 | 3.84±0.05 | 0.25±0.021 |
| DARTS-GAEA | 24.02±1.92 | **48.23±2.87** | 0.026±0.001 | **2.94±0.13** | 0.22±0.035 |
| DenseNAS | 25.98±0.38 | 72.99±0.95 | 0.10±0.01 | 3.84±0.15 | 0.38±0.038 |
| Auto-DL | - | - | 0.049±0.005 | 6.73±0.73 | 0.49±0.004 |
| BABY DASH | 25.56±1.37 | 63.45±0.88 | 0.016±0.002 | 3.94±0.54 | 0.16±0.007 |
| DASH | 24.37±0.81 | 71.28±0.68 | **0.0079±0.002** | 3.30±0.16 | 0.19±0.02 |

| | NinaPro 0-1 error (%) | FSD50K 1- mAP | ECG 1-F1 | Satellite 0-1 error (%) | DeepSEA 1-AUROC |
|---|---|---|---|---|---|
| WRN | 6.78±0.26 | 0.92±0.001 | 0.43±0.01 | 15.49±0.03 | 0.40±0.001 |
| TCN | - | - | 0.57±0.005 | 16.21±0.05 | 0.44±0.001 |
| Expert | 8.73±0.9 | 0.62±0.004 | **0.28±0.00** | 19.8±0.00 | 0.30±0.24 |
| Perceiver IO | 22.22±1.80 | 0.72±0.002 | 0.66±0.01 | 15.93±0.08 | 0.38±0.004 |
| WRN-ASHA | 7.34±0.76 | 0.91±0.03 | 0.43±0.01 | 15.84±0.52 | 0.41±0.002 |
| DARTS-GAEA | 17.67±1.39 | 0.94±0.02 | 0.34±0.01 | 12.51±0.24 | 0.36±0.02 |
| DenseNAS | 10.17±1.31 | 0.64±0.002 | 0.40±0.01 | 13.81±0.69 | 0.40±0.001 |
| AMBER | - | - | 0.67±0.015 | 12.97±0.07 | 0.68±0.01 |
| BABY DASH | 8.28±0.62 | 0.62±0.01 | 0.37±0.001 | 13.29±0.108 | 0.37±0.017 |
| DASH | **6.60±0.33** | **0.60±0.008** | 0.32±0.007 | **12.28±0.5** | **0.28±0.013** |

## A.7 Runtime of DASH on NAS-Bench-360

Table 6: Runtime breakdown for DASH on NAS-Bench-360 tasks evaluated on a NVIDIA V100 GPU.

| Task | Search | Hyperparameter Tuning | Retraining | Total |
|------|--------|----------------------|------------|-------|
| CIFAR-100 | 1.6 | 0.15 | 0.77 | 2.5 |
| Spherical | 1.6 | 0.25 | 3.16 | 5.0 |
| Darcy Flow | 0.16 | 1.6 | 3.5 | 5.3 |
| PSICOV | 0.88 | 0.64 | 14 | 15 |
| Cosmic | 1.6 | 0.055 | 5.1 | 6.8 |
| NinaPro | 0.028 | 0.16 | 0.11 | 0.30 |
| FSD50K | 0.88 | 0.88 | 27 | 29 |
| ECG | 0.18 | 0.28 | 0.83 | 1.3 |
| Satellite | 1.8 | 0.4 | 4.3 | 6.5 |
| DeepSEA | 0.36 | 1.6 | 8.3 | 10 |

## A.8 Searched Architecture Visualization

In this section, we give two example networks searched by DASH to show that large kernel matters for diverse tasks.

### A.8.1 2D Example: Darcy Flow

For this problem, DASH generates a WRN 16-4 [34] for retraining. The network architecture consists of several residual blocks. For instance, we can use $\text{Block}_{64,(7,1),(9,3)}$ to denote the residual block with the following structure:

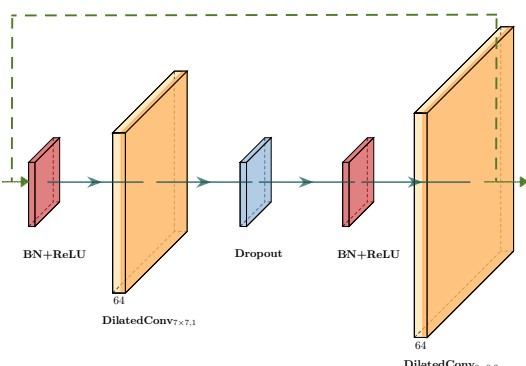

where $64$ is the output channel and BN denotes the BatchNorm layer. Note that size of a convolutional layer in the figure is proportional to the kernel size but not the number of channels. Then, an example network produced by DASH for Darcy Flow looks like the following:

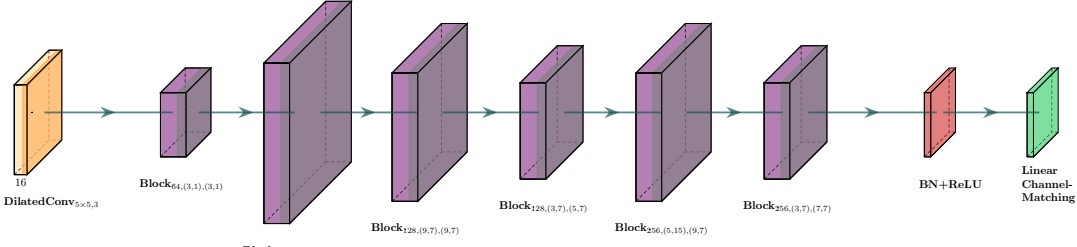

Since Darcy Flow is a dense prediction task, the last layer is a channel-matching (permutation+linear+permutation) layer instead of a pooling+linear layer for classification.

### A.8.2  1D Example: DeepSEA

For this problem, DASH generates a 1D WRN [35] for retraining. The network architecture consists of several residual blocks. For instance, we can use $\text{Block}_{64,(3,1),(5,3),(7,5)}$ to denote the residual block with the following structure:

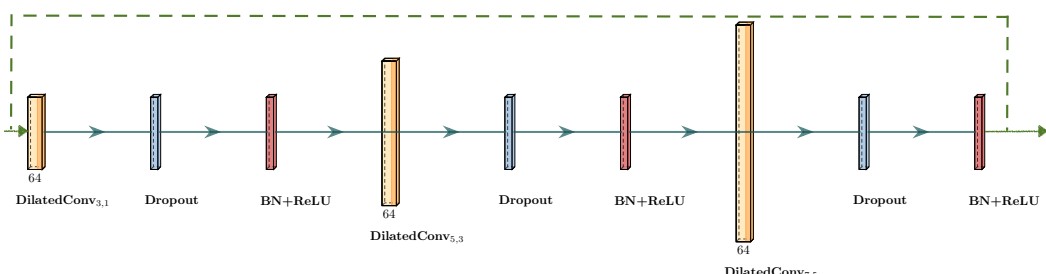

where $64$ is the output channel and BN denotes the BatchNorm layer. Then, an example network produced by DASH for DeepSEA looks like the following:

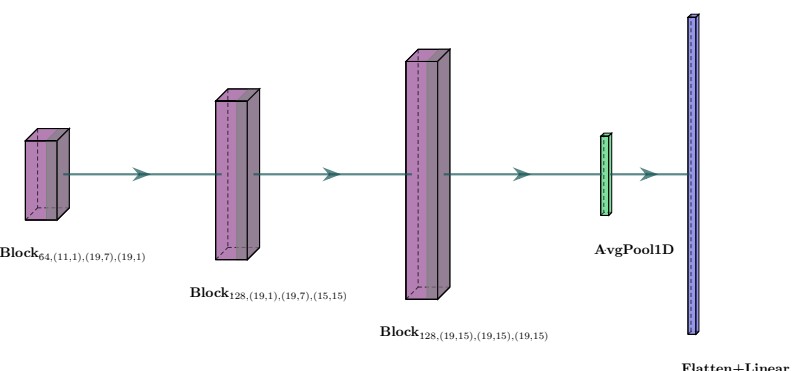

We can see that large kernels are indeed selected during search.

## A.9 Additional Results

### A.9.1 DASH-TCN for NAS-Bench-360

DASH works for all networks with a convolutional layer, so WRNs are not the only applicable backbone. Below, we provide the test errors of DASH with the 1D Temporal Convolutional Network backbone on some 1D tasks:

Table 7: Test errors for 1D NAS-Bench-360 tasks using the TCN backbone.

|  | ECG | Satellite | DeepSEA |
| --- | --- | --- | --- |
| Vanilla TCN | 0.57±0.005 | 16.21±0.05 | 0.44±0.001 |
| DASH-TCN | **0.29±0.004** | **12.39±0.043** | **0.24±0.012** |

We did not include the results in the paper to simplify presentation. Also, using WRNs in our workflow allows us to provide a fully automated pipeline that generates decent-performing models as quickly (due to its small size) and easily (due to the code for training WRNs being easily found online) as possible for previously unexplored tasks.

### A.9.2 DASH-ConvNeXt for ImageNet

Though our motivation is not to compete in the crowded vision domain but to provide a general solution to less-studied domains, we show that DASH is backward compatible with vision tasks by testing it on ImageNet-1K with two backbones of distinct scales. Our results show that DASH generalizes to tasks with large input shape ($3 \times 224 \times 224$), dataset size (1.2M), and number of classes (1000). It improves the accuracy of the original models and searches efficiently regardless of the backbone used.

We used Wide ResNet 16-4 (to be consistent with our workflow) and ConvNeXt-T [15] (a large-scale CNN that has onpar performance with SoTA Transformers) as the backbones and performed experiments on 4 NVIDIA V100 GPUs. To demonstrate DASH's efficiency, we first present the per-epoch search time (forward and backward time in secs) for three baselines over the search space $K = \{3, 5, 7, 9, 11\}$, $D = \{1, 3, 7\}$. A subset of 4096 images is used.

Table 8: Time for one search epoch (forward & backward) in seconds using different backbones.

|  | WRN | ConvNeXt |
| --- | --- | --- |
| # param | 3M | 28M |
| DASH | 151.3 | 80.5 |
| *Mixed-weights* | 705.4 | 300.1 |
| *Mixed-results* | 330.6 | 149.6 |

We can see that DASH's efficiency holds for both backbones. Though ConvNeXt has more parameters, it is searched faster than WRN as it has fewer conv layers and applies downsampling to the input.

Then, we report DASH's runtime vs. the train-time of the vanilla backbone (in hours). We let DASH search for 10 epochs with subsampling ratio 0.2. (Re)training takes 50 and 100 epochs for WRN and ConvNeXt, respectively.

Table 9: Runtime breakdown for DASH and the backbones on ImageNet-1K.

|  | WRN | ConvNeXt |
| --- | --- | --- |
| DASH search | 24 | 13 |
| DASH retrain | 52 | 48 |
| Backbone train | 16 | 41 |

Lastly, we report the top-1 accuracy of the searched vs. original models to show DASH generalizes to large vision input. We trained ConvNeXt for 300 epochs.

Table 10: Prediction errors (%) for DASH and backbones on ImageNet-1K. Backbone results are taken from [15].

|                      | WRN          | ConvNeXt     |
|----------------------|--------------|--------------|
| Vanilla Backbone     | 37.56±0.14   | 17.9±0.0     |
| DASH Searched Model   | **34.12±0.21** | **16.42±0.15** |

In general, DASH improves backbone performance by adopting task-specific kernels.