# OpenReview forum: "Efficient Architecture Search for Diverse Tasks"
_NeurIPS.cc/2022/Conference — NeurIPS 2022 Accept_

### Official Review · Reviewer_RXF5 · 2022-07-05

**Rating:** 6
**Confidence:** 3
**Soundness:** 2 fair
**Presentation:** 2 fair
**Contribution:** 2 fair

**Summary:**

This paper focuses on the direction of searching architectures for diverse tasks. The designed search space contains convolution layers with large and diverse set of kernel sizes and dilation rates. To address the efficiency challenges encountered in weight-sharing, DASH is proposed by using mixture-of-operations based on the Fourier diagonalization of convolution. Extensive experiments on ten tasks suggest the effectiveness of the proposed DASH methods.

**Questions:**

1. I 'm wondering the performance of DASH in the designed search space with convolutions of large kernel sizes and dilation sizes in large data sets, like ImageNet.
2. The other two questions are listed in the weakness part.
Looking forward to authors' responses.

**Limitations:**

Yes.

**Strengths And Weaknesses:**

Strengths:
1. This paper explores a new direction of applying NAS in diverse tasks. In their setting, the authors notice the importance of involving convolution layers with large kernel sizes and dilation sizes. To address the efficiency problem in the large search space, mixture-of-operations based on the Fourier diagonalization of convolution is used in their proposed DASH.
2. Extensive experiments validate the importance of involving convolution operations with large kernel sizes and the effectiveness of DASH in ten tasks.

Weaknesses:
1. For diverse tasks, one important setting is that whether we can directly search architectures from the super-network for different tasks. It seems that the authors did not take this into account.
2. In section 3.3, grid search is used to search hyper-parameters in both search and retraining stage. Does grid search play a key role in performance? If grid search is also applied to baseline methods, will it improve the performance?

---

> ### Author Response · Authors · 2022-08-02
> **Response to Reviewer RXF5**
>
> Thank you for your positive feedback. We hope to respond to your questions below.
>
> **_1. [One important setting is whether we can directly search architectures from the super-network for different tasks. It seems that the authors did not take this into account.]_**
>
> We are not quite sure what you mean by this comment, though we would be happy to discuss more during the discussion phase. To clarify, we indeed use super-networks to find architectures for different tasks. Specifically, a super-network is generated by 1) selecting a backbone, and 2) replacing the backbone's conv layers with the aggregated conv operation which expresses a set of kernels of varying sizes. The architecture parameters are then optimized. The final obtained model will be a sub-network in the sense that the conv operation selected at each layer is one of the predefined kernels.
>
> On the other hand, if you are suggesting using the same super-network to simultaneously search for architectures for different tasks, it would be helpful to have references for this being somehow standard. If you are referring to methods like Once-For-All [1], then note that their focus is on specializing architectures from a single super-network trained on one task for *diverse devices*, whereas we consider diverse tasks with varying input and output dimensions and types. We are not aware of many NAS papers (outside of low-data settings such as meta-learning) that simultaneously train super-networks on multiple tasks with different datasets.
>
> **_2. [Does grid search play a key role in performance? If grid search is also applied to baseline methods, will it improve the performance?]_**
>
> The role of automated hyperparameter tuning in DASH is to stabilize *retraining* rather than to significantly improve performance. It also does not affect search. For better comparison, we include the WRN-ASHA baseline which uses the same hyperparameter tuning technique but with a much larger search space and tuning budget. However, DASH significantly outperforms WRN-ASHA, showing that simply tuning the hyperparameters of the baseline is insufficient. Learning the kernel sizes from the data is important.
>
> Below is a more detailed answer to your question.
> - **Why do we need this step?** Grid search is added because DASH may favor large kernels when the task requires modeling long-range dependencies, and previous work [2, 3] has shown that optimizing large kernels often requires smaller learning rates or higher regularization. Grid search consumes a small portion of the runtime (Appendix Table 6) so it can be added at a low cost to stabilize retraining.
> - **Does grid search play a key role in performance?** During the development of DASH, we have evaluated DASH *without* hyperparameter tuning and gathered the following results:
> | |CIFAR-100 (l)|Spherical (l)|Darcy Flow (l)|PSICOV (l)|Cosmic (l)|NinaPro (l)|ECG (h)|Satellite (l)|DeepSEA (h)|
> |:-:|:-:|:-:|:-:|:-:|:-:|:-:|:-:|:-:|:-:|
> |DASH with hp tuning|24.37|75.44|0.0079|3.30|15.04|6.6|0.4|0.68|12.28|0.72|
> |DASH without hp tuning|25.36|75.4|0.008|3.34|21.06|5.85|0.48|0.71|12.39|0.76|
>
> (l/h indicates lower/higher is better; results are averaged over 3 trials.)
>
> Note that DASH without grid search already outperforms all competitors on 6/10 tasks, and lower learning rates indeed help with tasks such as Cosmic and Satellite. However, as the naive grid search can be suboptimal, the tuned hyperparameters can perform worse than the fixed ones in tasks such as ECG. Still, we kept hyperparameter tuning in DASH to ensure robustness, so that DASH does not have failure modes for previously unseen tasks. We will add the above discussion to the paper.
>
> **_3. [I 'm wondering the performance of DASH in the designed search space with convolutions of large kernel sizes and dilation sizes in large data sets, like ImageNet.]_**
>
> Please refer to our general response for ImageNet1K results, which shows that DASH is applicable to large vision datasets and can improve the performance of existing models.
>
> In the new experiments, we included kernels with max size 11 and dilation rate 7, so the max receptive field size is 7x(11-1)+1=71, which is significantly larger than the 3x3 kernel used by common vision CNNs. As is also pointed out in the general response, DASH indeed selects kernels with larger sizes. This implies that large kernels are preferable for extracting higher-level features. Even with such a large search space and a large ConvNeXt backbone (45M params, 4.5G FLOPS), the time for one search epoch with DASH is only 3x that of one training epoch with the fixed-kernel backbone. This backs up our claims about DASH efficiency.
>
> **_References_**
>
> [1] Cai, Gan et al. Once-for-All: Train one network and specialize it for efficient deployment. ICLR 2020.
>
> [2] Liu, Zhuang et al. A ConvNet for the 2020s. CVPR 2022.
>
> [3] Ding, Xiaohan et al. Scaling Up Your Kernels to 31x31: Revisiting Large Kernel Design in CNNs. CVPR 2022.

---

> > ### Comment · Reviewer_RXF5 · 2022-08-06
> > **Response after the rebuttal**
> >
> > Thanks for the rebuttal and most of my concerns are resolved. But I still have the concern about the usefulness of the proposed DASH in simultaneously searching the architectures from the same super-network, as nowadays applications (like self-driving) require one backbone to serve for different tasks (e.g., detection, segmentation).

---

> > > ### Author Response · Authors · 2022-08-06
> > > **Response to New Application Scenarios**
> > >
> > > Thank you for your response. We would like to make the following high-level points:
> > > 1. The use case you mentioned—simultaneously obtaining architectures from one super-network for different tasks in different domains (if we understand it correctly; if not, references would be helpful)—is an important but to-date open question. However,  we view this as clearly being future work, and would like to remind the reviewer that DASH has already made significant strides, as evidenced by its superior performance relative to NAS and expert baselines for various diverse tasks, especially those beyond the intensively studied vision domain.
> > > 2. Based on our reading of your comment, we are thus surprised that your current review score seems to be largely focused on this particular use case and applications such as self-driving. Given that DASH has already been thoroughly evaluated, that it explicitly aims to tackle tasks beyond vision, and that we appear to have addressed all of your other concerns, we would like to ask the reviewer to consider raising their score.

---

> > > ### Author Response · Authors · 2022-08-09
> > > **Updated ImageNet and ConvNeXt Results**
> > >
> > > We have updated the DASH-ConvNeXt performance results on ImageNet in our general response. Since the discussion period is drawing to a close, we hope that our responses have provided you with enough information to address any concerns you may have. Please let us know if there are any further clarifications that we can provide. We appreciate your time and effort in giving valuable feedback on our work.

---

### Official Review · Reviewer_APUY · 2022-07-12

**Rating:** 5
**Confidence:** 3
**Soundness:** 2 fair
**Presentation:** 2 fair
**Contribution:** 2 fair

**Summary:**

This paper introduces a NAS method called NASH. This method can be applied to a variety of tasks and achieve good results. It is meaningful that NAS is applied to a variety of tasks, which demonstrates the potential of NAS methods.

**Questions:**

Please refer to the weakness in the above part.

**Limitations:**

Please refer to the weakness in the above part.

**Strengths And Weaknesses:**

# Strengths
+ The tricks used for acceleration are a bit interesting.
+ It makes sense to apply the NAS method to other tasks.

# Weaknesses
+ Table3 is mentioned in many places, but I didn't find it.
+ There are some problems with the experimental part：
1. The FLOPs for each model are not reported, and it is not clear to me whether these methods are comparable.
2.  Is the search space the same for each model?
3. Why only use WRN for experiments? Different structures should be chosen to verify the effectiveness of the method.

+ Frankly speaking, NASH is not a surprising method in terms of method innovation, it is just an application of DARTS to some extent.

---

> ### Author Response · Authors · 2022-08-02
> **Response to Reviewer APUY**
>
> Thank you for your feedback. We respectfully but strongly disagree with the weaknesses you discuss. In particular, you raised a few issues about a missing table, incomparable baselines, and a lack of methodological novelty. These claims are either incorrect or misplaced, as we describe below.
>
> **_1. [Table 3 is mentioned in many places, but I didn't find it.]_**
>
> Table 3 is on the bottom-right corner of page 8. It shows that DASH is consistently faster than DARTS, and the combined search and retraining time is less than a factor of two slower than training a backbone on most tasks.
>
> **_2. [The FLOPs for each model are not reported, it is not clear to me whether these methods are comparable.]_**
>
> Following the framework of NAS-Bench-360, the computational efficiency metric we use is the time to search for an architecture and train a final model. DASH is much more efficient than the closest competitor-in-accuracy, DARTS, on all tasks. The FLOPS of the searched models are difficult to compare because different NAS algorithms use different search strategies, so the resulting models have drastically different structures—DARTS, for example, builds models by repeatedly stacking the same searched cell, whereas our method modifies the kernels in the backbone and allows layer operations to vary from the beginning to the end of a network.
>
> In the new ImageNet experiments, we measured the FLOPs of applying DASH and mixed-results (applying DARTS to our search space) to the backbones using the PyTorch library `ptflops`. It is evident that DASH significantly reduces floating-point computations.
> |FLOPs (GMac)|WRN|ConvNeXt|
> |:-:|:-:|:-:|
> |DASH|1.65|1.58|
> |Mixed-results|45.48|15.75|
>
> **_3. [Is the search space the same for each model?]_**
>
> Across different baselines, e.g., DARTS, DenseNAS, and DASH, the search spaces are different. Since our main contribution is a new search space, it needs to be compared with different search spaces already in use to show its usefulness. Besides, different NAS methods generate models in distinct ways (as described in the response to question 2), so the search spaces are naturally different.
>
> Across different tasks that DASH is applied to, the search spaces KxD are specified in Section A.5.2. They are different but are completely determined by the input shape (as described in Section 3.3) to make our workflow as automated as possible.
>
> **_4. [Why only use WRN for experiments? Different structures should be chosen to verify the effectiveness of the method.]_**
>
> DASH works for all networks with a conv layer, so you are correct that WRN is not the only applicable backbone. In fact, when developing DASH, we tested it on 1D Temporal Convolutional Network:
> | |ECG (h)|Satellite (l)|DeepSEA (h)|
> |:-:|:-:|:-:|:-:|
> |Vanilla TCN|0.43|16.21|0.56|
> |DASH TCN|0.71|12.39|0.76|
>
> (l/h indicates lower/higher is better; results are averaged over 3 trails.)
>
> We did not include the results in the paper to simplify presentation, so that all of our backbones are ResNet-like. Also, using WRNs in our workflow allows us to provide a fully automated pipeline that generates decent-performing models as quickly (due to its small size) and easily (due to the code for training WRNs being easily found online) as possible for unseen tasks.
>
> In addition to TCN, we also show results for evaluating DASH with ConvNeXt on ImageNet in the general response, which again demonstrates that DASH works for different backbones and datasets. We agree that evaluating different structures is useful, so we will add the above results to the revision.
>
>
> **_5. [NASH is not a surprising method in terms of method innovation, it is just an application of DARTS to some extent.]_**
>
> We believe this is an unfair and reductive characterization of the work. The main contribution of DARTS is a new search algorithm, whereas ours is a new search space. More importantly, DASH is not an application of DARTS because DARTS is *insufficient* to explore the large and diverse (e.g. k=19, d=15) kernel space that we propose. The continuous relaxation scheme of DARTS on its own is slow and only applicable to small search spaces with small kernels (e.g. k=5, d=2). Theoretical and empirical evidence can be found in Table 1, Figs. 2 and 3.
>
> Our focus on diverse tasks leads us to propose a search space that is both natural (containing many kernels for multi-scale feature extraction) and nontrivial (existing techniques cannot efficiently search it). To improve search efficiency, we make technical contributions on top of continuous relaxation and obtain strong performance on an extensive benchmark. Thus, DASH has technical novelty and practical impact. This is echoed by reviewer nUHZ: *“although essential the ideas in this paper are based on several earlier works (e.g. DARTS or [Wu et al:Tree structured network...])...the recognition that reusing some of the time-consuming intermediate results make these methods much more appealing in this context is novel and useful.”*

---

> ### Author Response · Authors · 2022-08-07
> **Looking Forward to Further Feedback**
>
> We appreciate the reviewer's time and effort in providing valuable suggestions for our work. Please let us know if our response has addressed your concerns. We would be happy to answer any other questions you may have.

---

> ### Author Response · Authors · 2022-08-09
> **Updated ImageNet and ConvNeXt Results**
>
> We have updated the DASH-ConvNeXt performance results on ImageNet in our general response. Since the discussion period is drawing to a close, we hope that our responses have provided you with enough information to address any concerns you may have. Please let us know if there are any further clarifications that we can provide. We appreciate your time and effort in giving valuable feedback on our work.

---

### Official Review · Reviewer_nUHZ · 2022-07-19

**Rating:** 6
**Confidence:** 4
**Soundness:** 3 good
**Presentation:** 2 fair
**Contribution:** 3 good

**Summary:**

This paper gives a significantly improved neural architecture discovery method specialized for convolutional networks. The idea is to train a large super-network and cache common computation and select the best performing subnetwork. The paper also presents several useful ideas to accelerate to training and exploration of convolutional operators at a large scale. This is based on the observation that the overhead of using Fourier transformation is significantly amortized if the transformation is used for exploring convolutions for a wider range of hyperparameters including patch size and dilation factor. In addition the work suggests exploration on multiple tasks at the same tasks for more robust results.

**Questions:**

Could you give more experimental evidence on benchmarks with large images and more classes (esp ImageNet) and other vision tasks.


**Limitations:**

I don't see any concerns regarding the societal impact specific to this work.

**Strengths And Weaknesses:**

Originality: High

Although essential the ideas in this paper are based on several earlier works (e.g. DARTS or [Wu et al:Tree structured network...]) or the use of Fourier transformations has ben extensively explored in the literature. However the recognition that reusing some of the time-consuming intermediate results make these methods much more appealing in this context is novel and useful.

Quality: Mixed.

The methods are well-motivated and seemingly efficient, but the experimental results lack some more thorough evidence, especially on images of larger size. While the synergy of the applied methods is very high and seems to result in a very efficient methodology, the search space is limited to a very specific type of convolutional deep learning models. The experimental section also lacks more thorough ablation for judging the criticality and

Clarity: Medium.

The description of the applied methods is extremely formal/mathematical and would benefit from a more intuitive explanations and more figures to introduce the later to the most essential ideas in a more gentle way. Also, the paper is not structured in a very appealing manner and could have been made more accessible by a better top-down explanation which would explain the synergy of the components and more thorough intuitive comparisons with similar approaches.

Significance: Medium. Given the fact that transformers are overtaking convolutional networks in many respects, it is hard to judge the future significance of these ideas as transformer-based methods, or even just using transformer-based exploration might render these methods less appealing in the near future. Also the lack of experimental results on benchmarks with larger images (like ImageNet) is concerning and makes it harder to judge the expected impact of the methods.

---

> ### Author Response · Authors · 2022-08-02
> **Response to Reviewer nUHZ**
>
> Thank you for your positive review. We hope to address your questions below.
>
> **_1. [the experimental results lack more thorough evidence, especially on images of larger size... the search space is limited to a very specific type of convolutional deep learning models. The experiment section also lacks a more thorough ablation.]_**
>
> Please note that our main evaluation is on a benchmark of 10 different datasets, which we (and Reviewer RXF5) view as “extensive.” As for results on larger images, please see our general response for ImageNet evaluation, which shows that DASH generalizes to large vision input. We also tested DASH on different backbones such as ConvNeXt and TCN (see response to reviewer APUY, question 4), which shows that it can improve the accuracy of different models. Note that we have performed ablation studies for the necessity of search spaces with large kernels and dilations (via the comparison to BABY DASH) and for the acceleration techniques to handle the larger search space (via Figs. 2 and 3 in the paper, and the table for ImageNet search time).
>
> **_2. [Given the fact that transformers are overtaking convolutional networks in many respects, it is hard to judge the future significance of these ideas as transformer-based methods, or even just using transformer-based exploration might render these methods less appealing.]_**
>
> We agree that Transformers are gaining popularity in a number of domains, but we do not think this trend will render our method obsolete for the following reasons:
>
> 1. Many Transformer models still depend on convolutions for feature extraction and transformation, and their performance relies on the quality of the embedded features. Since DASH is applicable to any architecture with a conv layer, it can be helpful for many models, including Vision Transformer with a conv patching layer [1], Deformable Transformer with a ResNet embedder [2], Swin Transformer with a conv decoder [3], and many others.
> 2. Recent work like ConvNeXt [4] suggests that CNNs can take advantage of many techniques used by Transformers to match their performance on large-scale datasets without using the attention mechanism. Thus, there is reason to believe that CNNs will remain competitive in the future. Furthermore, DASH may help in this regard, as one of the tricks that ConvNeXt needs is the use of large kernel sizes.
> 3. The goal of this paper is to develop an automated and general method for diverse-task solving. We believe that DASH is a crucial *first step*, in the sense that no existing method that we know of can achieve the same level of generalizability. In particular, we have collected important evidence that current Transformer-based methods still have difficulty solving diverse tasks in both of the following settings:
>     - Train from scratch: the Perceiver IO baseline in Table 2 performs poorly on many tasks.
>     - Pretrain & fine-tune: after submission, we fine-tuned 4 models on NAS-Bench-360 without modifying the architectures. These models perform well on in-domain tasks (e.g. ViT on CIFAR) but poorly on out-of-domain ones. In contrast, DASH’s generalizability and convolution’s capacity as diverse feature extractors are demonstrated by the performance profile (Fig. 4) in our paper.
> | |CIFAR-100 (l)|Spherical (l)|Darcy Flow (l)|PSICOV (l)|Cosmic (l)|NinaPro (l)|ECG (h)|Satellite (l)|DeepSEA (h)|
> |:-:|:-:|:-:|:-:|:-:|:-:|:-:|:-:|:-:|:-:|
> |ViT (2D Point)|8.38|49.34|-|-|-|11.56|-|-|-|
> |DETR (2D Dense)|-|-|0.011|3.45|29.95|-|-|-|-|
> |RoBERTa (1D)|-|-|-|-|-|-|0.68|13.46|0.64|
> |wav2vec (1D)|-|-|-|-|-|-|0.61|15.68|0.61|
>
> (l/h indicates lower/higher is better.)
>
> Meanwhile, DASH is flexible because users have the freedom to choose the backbone based on available computational resources. E.g., if the goal is to deploy a model on mobile devices, one can use a smaller backbone like ResNet-20. We have included instructions for integrating DASH with any model that has a conv layer in our code.
>
> **_3. [The paper is not structured in a very appealing manner and could have been more accessible by a top-down explanation which would explain the synergy of the components and more intuitive comparisons with similar approaches.]_**
>
> Thank you very much for the new perspective on paper organization. We will add more intuitive explanations to the paper.
>
> **_References_**
>
> [1] Dosovitskiy, Alexey et al. An Image is Worth 16x16 Words: Transformers for Image Recognition at Scale. ICLR 2021.
>
> [2] Carion, Nicolas et al. End-to-End Object Detection with Transformers. 2020.
>
> [3] Liu, Ze et al. Swin Transformer: Hierarchical Vision Transformer using Shifted Windows. ICCV 2021.
>
> [4] Liu, Zhuang et al. A ConvNet for the 2020s. CVPR 2022.

---

> ### Author Response · Authors · 2022-08-07
> **Looking Forward to Further Feedback**
>
> We appreciate the reviewer's time and effort in providing valuable suggestions for our work. Please let us know if our response has addressed your concerns. We would be happy to answer any other questions you may have.

---

> ### Author Response · Authors · 2022-08-09
> **Updated ImageNet and ConvNeXt Results**
>
> We have updated the DASH-ConvNeXt performance results on ImageNet in our general response. Since the discussion period is drawing to a close, we hope that our responses have provided you with enough information to address any concerns you may have. Please let us know if there are any further clarifications that we can provide. We appreciate your time and effort in giving valuable feedback on our work.

---

### Author Response · Authors · 2022-08-02
**General Response & ImageNet1K Results**

We thank the reviewers for their helpful feedback. Here, we briefly summarize our contributions and present evaluations of DASH on a large-scale dataset (ImageNet1K) and a new backbone (ConvNeXt-T). The results show that our claims about DASH’s efficiency and effectiveness extend beyond the tasks and backbones in the original submission, directly addressing concerns raised by the reviewers.

In this work, we first identified a crucial problem in today’s ML research: while many models are being developed in popular domains like CV and NLP, these models only work for the specific tasks they are trained on and often cannot generalize to distinct fields. Thus, selecting the right models for arbitrary tasks in understudied domains remains difficult. A key to solving this issue is to automatically design networks that capture input features based on a task's needs—while some tasks require modeling neighboring interactions, others require modeling long-range dependencies. To achieve such task-specific feature extraction, we present DASH, which customizes the convolutional operations in a model by searching for the optimal kernel sizes and dilation rates in a large and diverse search space. DASH represents a new framework for model development in diverse domains with the following benefits:
- **Efficiency**: DASH explores the large kernel space efficiently via a novel combination of weight-mixing, Fourier convolution, and Kronecker dilation, whereas existing NAS methods cannot do so.
- **Effectiveness**: DASH finds optimal kernels for a wide set of problems. On more than 6 evaluated tasks, the searched models beat all NAS baselines and hand-designed expert models.
- **Generalizability**: DASH uses an aggregated conv operator to extract features at different granularities. Its ability to generate large-kernel networks allows it to tackle new tasks like PDE solving and protein folding. Existing NAS and general-purpose methods all have failure modes due to their limited sets of feature extractors considered.

Besides these novelties, DASH has practical significance as it can be applied to any model with a conv layer, such as Vision Transformer [1], CoAtNet [2], to enhance model performance at a low cost.

## ImageNet1K Results
All reviewers mentioned testing DASH on larger image datasets and more backbones. Though our motivation is not to compete in the crowded vision domain but to provide a general solution to less-studied domains, we agree that showing “backward compatibility” is crucial. To complement existing evaluations, we tested DASH on ImageNet1K with two backbones of distinct scales. Results show that DASH generalizes to tasks with large input shape (3x224x224), dataset size (1.2M), and number of classes (1000). It improves the accuracy of the original models and searches efficiently regardless of the backbone used.

We used Wide ResNet 16-4 (to be consistent with the AutoML workflow in the paper) and ConvNeXt-T (a large-scale CNN that has onpar performance with SoTA Transformers) as the backbones and performed experiments on 4 NVIDIA V100 GPUs. To demonstrate DASH's efficiency, we first present the per-epoch search time for three baselines over the search space K={3,5,7,9,11}, D={1,3,7}. A subset of 4096 images is used.
|Search time (secs/epoch)|WRN|ConvNeXt|
|:-:|:-:|:-:|
|# params|3M|28M|
|DASH|151.3|80.5|
|Mixed-weights|705.4|300.1|
|Mixed-results|330.6|149.6|

DASH's efficiency holds for both backbones. Though ConvNeXt has more parameters, it is searched faster than WRN as it has fewer conv layers and applies downsampling to the input.

Then, we report DASH's runtime vs. the train-time of the vanilla backbone. We let DASH search for 10 epochs with subsampling ratio 0.2. (Re)training takes 50 and 100 epochs for WRN and ConvNeXt, respectively.
|Total time (hrs)|WRN|ConvNeXt|
|:-:|:-:|:-:|
|DASH search|24|13|
|DASH retrain|52|48|
|Backbone train|16|41|

Lastly, we report the top-1 accuracy of the searched vs. original models to show DASH generalizes to large vision input. Note that due to the limited response time, we trained ConvNeXt for 100 epochs and a single trial. In the revision, we will include results with full training (300 epochs as in [3]) and more trials.
|Acc|WRN|ConvNeXt|
|:-:|:-:|:-:|
|Vanilla backbone|60.1|76.3|
|DASH searched model|64.5|78.4|

In general, DASH improves backbone performance by adopting task-specific kernels. We observe that it tends to pick larger kernels for later layers, possibly to extract higher-level features. E.g., a set of kernels (k, d) it selects for WRN is (3,3)&rarr;(7,1)&rarr;(7,3)&rarr;(11,3)&rarr;(7,3)&rarr;(11,1)&rarr;(7,3)&rarr;(11,1)&rarr;(7,1)&rarr;(11,7)&rarr;(7,3)&rarr;(11,3)&rarr;(11,1).

**_References_**

[1] Dosovitskiy, Alexey et al. Transformers for Image Recognition at Scale. ICLR 2021.

[2] Dai, Zihang et al. CoAtNet: Marrying Convolution and Attention for All Data Sizes. NeurIPS 2021.

[3] Liu, Zhuang et al. A ConvNet for the 2020s. CVPR 2022.

---

> ### Author Response · Authors · 2022-08-09
> **Updated DASH-ConvNeXt Full Training Results on ImageNet1K**
>
> We wanted to provide an update on our ImageNet1K experiments. We have trained the DASH-searched models for a longer time (300 epochs as specified in the ConvNeXt paper) and obtained the following top-1 accuracy:
>
> |Acc (300-epoch)|WRN 16-4|ConvNeXt-T|
> |:-:|:-:|:-:|
> |Vanilla backbone|62.2|82.1|
> |DASH-searched model|66.6|83.5|
>
> Though it is only one trial so far, we believe a 1.4% increase on top of the ConvNeXt backbone, which was developed in part via manual tuning of the kernel size, is a significant improvement. These results provide further support that DASH is backbone-agnostic, and it can be used to augment modern architectures with task-specific kernels to solve diverse problems effectively and efficiently.
>
> We will add the experiments with ImageNet and ConvNeXt to the paper and use more random seeds. Since the discussion period is drawing to a close, we hope that our responses have provided the reviewers with enough information to address their concerns. Please let us know if there are any further clarifications that we can provide.

---

### Meta-Review · Area_Chair_DBVH · 2022-08-27

**Recommendation:** Accept
**Confidence:** Certain

**Metareview:**

This paper proposes a significantly improved neural architecture discovery method specialized for convolutional networks.

The idea is to train a large super-network and cache common computation and select the best performing subnetwork.

Also, it presents several useful ideas to accelerate to training and exploration of convolutional operators at a large scale.

While the work is the combination of somewhat well-known approaches, the overall approach is novel, interesting and well-motivated, however its application domain is somewhat limited.

Still, given the convincing experimental results, good execution and significant gains in terms of training time compared to competing appraoches, I propose this paper to be accepted for NeurIPS 2022.

**Award:**

No

---

### Decision · Program_Chairs · 2022-09-14

Accept